# Random forest vs. logistic regression: Predicting angiographic in-stent restenosis after second-generation drug-eluting stent implantation

Zhi Jiang[1,2☯], Longhai Tian[1,2☯], Wei Liu[1,2], Bo Song[1,2], Chao Xue[1,2], Tianzong Li[1,2], Jin Chen[1,2], Fang Wei[1,2]*

1 Cardiology Department, Guizhou Provincial People's Hospital, Guiyang, China, 2 Guizhou Provincial Cardiovascular Disease Institute, Guiyang, China

☯ These authors contributed equally to this work.
* wf13639087323@163.com

## Abstract

As the rate of percutaneous coronary intervention increases, in-stent restenosis (ISR) has become a burden. Random forest (RF) could be superior to logistic regression (LR) for predicting ISR due to its robustness. We developed an RF model and compared its performance with the LR one for predicting ISR. We retrospectively included 1501 patients (age: 64.0 ± 10.3; male: 76.7%; ISR events: 279) who underwent coronary angiography at 9 to 18 months after implantation of 2$^{nd}$ generation drug-eluting stents. The data were randomly split into a pair of train and test datasets for model development and validation with 50 repeats. The predictive performance was assessed by the area under the curve (AUC) of the receiver operating characteristic (ROC). The RF models predicted ISR with larger AUC-ROCs of 0.829 ± 0.025 compared to 0.784 ± 0.027 of the LR models. The difference was statistically significant in 29 of the 50 repeats. The RF and LR models had similar sensitivity using the same cutoff threshold, but the specificity was significantly higher in the RF models, reducing 25% of the false positives. By removing the high leverage outliers, the LR models had comparable AUC-ROC to the RF models. Compared to the LR, the RF was more robust and significantly improved the performance for predicting ISR. It could cost-effectively identify patients with high ISR risk and help the clinical decision of coronary stenting.

## Introduction

Percutaneous coronary intervention (PCI) has been a routine clinical practice for revascularization in patients with coronary artery disease (CAD), by reducing mortality in ST-segment elevation myocardial infarction and improving quality of life [1]. The mid-term risk of death associated with PCI using second-generation drug-eluting stent (DES) is close to that associated with coronary artery bypass grafting, except for individuals with diabetes and/or three-vessel disease [2]. As the rate of revascularization by stenting continues to increase, in-stent

**Data Availability Statement:** Minimal data set was uploaded as Supporting Information files.

**Funding:** FW recieved the Clinical Research Center Project of Department of Science and Technology

of Guizhou Province [NO.(2017)5405]; ZJ recieved the Guizhou Provincial High-level Innovative Talents Project (GZSYQCC[2015]006); WL recieved the Guizhou Provincial Science and Technology Foundation (No.[2019]1197); FW recieved the Guizhou Provincial Science and Technology Social Development Project (No. [2018]2794). The funders had no role in study design, data collection and analysis, decision to publish, or preparation of the manuscript.

**Competing interests:** The authors have declared that no competing interests exist.

restenosis (ISR) has become a burden that impairs patient well-being [3]. About 10% of the PCIs in the United States were for ISR lesions, and approximately 25% of the patients with ISR presented acute myocardial infarction. Predicting ISR would enable the possibility to optimize stent procedure, closely monitor or consider an alternate treatment. However, the existing risk model has not been used in clinical practice due to limited external validation [4]. A powerful and robust prediction model is urgently needed.

Logistic regression (LR) is a standard approach for binary prediction, but it is easily impacted by outliers [5]. Outlier is an observation point that is distant from other observation points. They produce leverage effect to the LR model and impair its predictive performance. Recently, random forest (RF), a machine-learning (ML) algorithm, has gained popularity in predicting clinical outcomes. In a large-scale benchmark experiment, RF outperformed LR in prediction in 69% of datasets from open ML databases [6]. Vien et al. found that the RF model was superior to the traditional LR model in predicting pacemaker implantation following transcatheter aortic valve replacement [7]. A previous study reported that the ML-based algorithms had higher accuracy than the existing risk score model in predicting ISR [8]. But no significant difference was revealed between RF and LR due to the small sample size (263 patients with 23 ISR events).

We hypothesized that the RF model can be used in ISR prediction and have better performance than the LR model due to higher robustness. We developed an RF model and compared its predictive metrics to the LR model in a larger retrospective dataset including 1501 patients and 279 ISR events. The robustness of the RF and LR model was also tested. The following article is presented following the STROBE reporting checklist [9].

## Method

### Data source

The data were retrospectively collected from the database at Guizhou Provincial People's Hospital with the institutional ethics committee's approval. The requirement for informed participant consent was waived by the ethics committee since the data were deidentified. ISR is defined as more than 50% stenosis within or 5 mm adjacent to a previously stented segment by quantitative coronary analysis (QCA; syngo QCA software, Siemens) [10]. We screened 2508 patients who had reassessed coronary dimension by QCA within 9 to 18 months after prior coronary stenting between January 2014 and August 2020 (Fig 1). We excluded 967 patients who had prior stenting procedures in other hospitals, 16 receiving stents other than 2nd generation DES, and 24 with missing essential clinical characteristics. A total of 1501 patients were finally included in the study. 279 patients were diagnosed with ISR. 1222 patients without ISR were identified as control. No patients with PCI in bypass grafts were included.

### Model development

We used the open-source R software version 4.0.5 (The R Foundation, Vienna, Austria) for ML model development. The data were randomly split into the train (75%) and test datasets (25%) with 50 repeats, generating 50 paired train and test datasets (Fig 1). Each test dataset was unseen to its paired train dataset. The 10-fold cross-validation method was used for tuning hyperparameters, selecting variable subsets, and developing models in the train datasets. Then each model was validated in the paired test dataset. The design was aimed to avoid any over-optimistic or over-pessimistic results by chance.

15 ISR predictors were selected according to documentation, including patient age, male gender, smoking history, clinical presentation of acute coronary syndrome (ACS), diabetes, hypertension, dyslipidemia, chronic kidney disease (CKD) stage, number of stenotic vessels

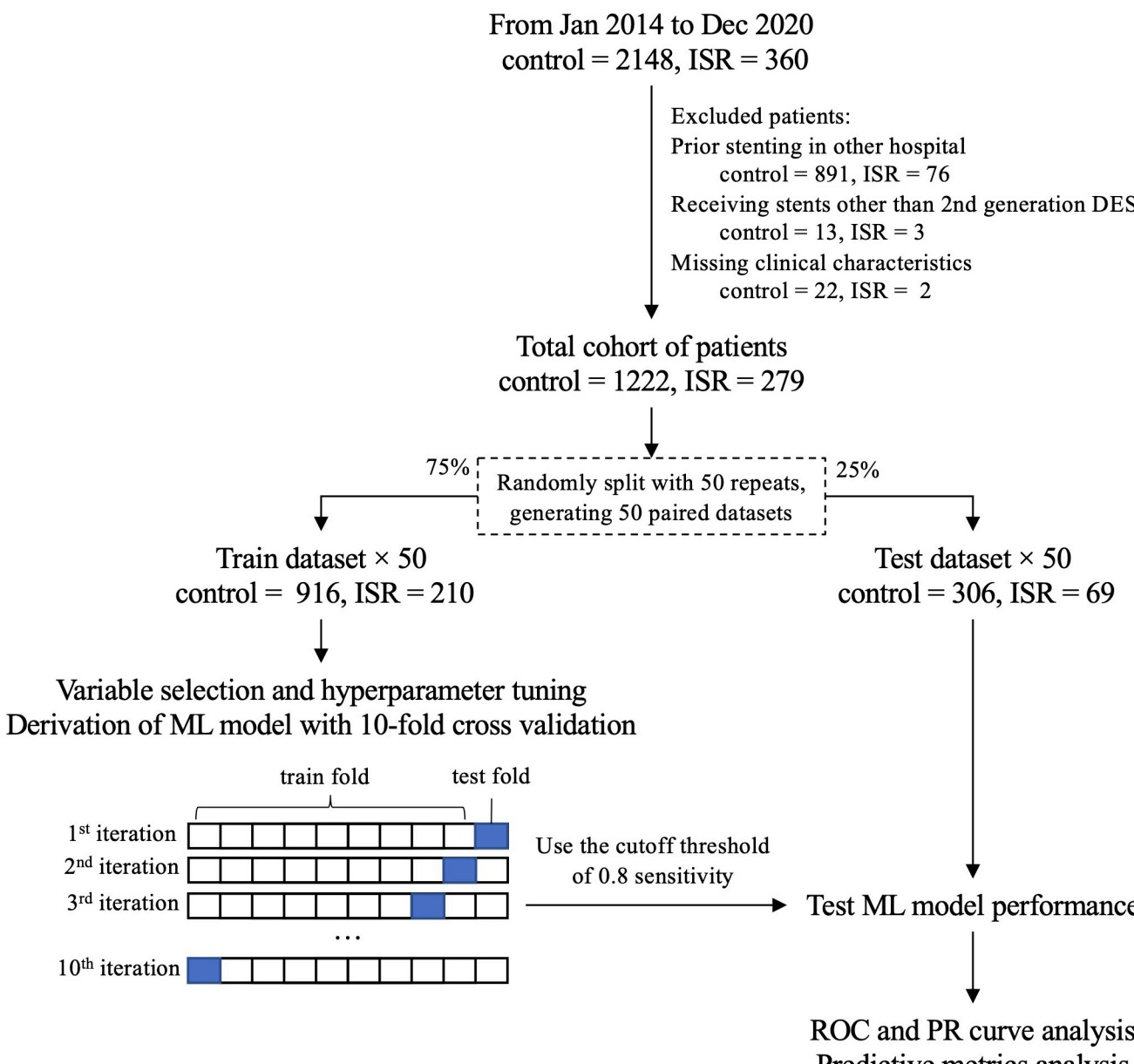

**Fig 1. Overview of the data source and model development. Abbreviations:** DES = drug-eluting stent, ISR = intra-stent restenosis, ROC = receiver operating characteristic, PR = precision-recall.

(>50% of luminal diameter by QCA), number of stenting vessels, minimum stent diameter, total stent length, left main artery stenting, bifurcation stenting (left main), and stenting in complex lesions (type B2 and C) [11]. The CKD stage is classified by calculating the estimated glomerular filtration rate using the Modification of Diet in Renal Disease Study equation [12]. The complex lesion was classified by two experienced interventional cardiologists according to the ACC/AHA criteria [13].

The hyperparameter of mtry (the number of random feature candidates at each split) and ntree (the number of trees in forest) in the RF model were first tuned by the grid search method (caret package, version 6.0–88; randomForest package, version 4.6–14). Then we

measured the conditional permutation importance (CPI), mean decrease accuracy (MDA), and mean decrease Gini (MDGini) to select important variables using the 50 train datasets (permimp package, version 1.0–1) [14]. We unselected left main artery stenting and bifurcation stenting due to their low values in all the parameters (S1 Fig). For the LR model, the stepwise Akaike information criterion (AIC) method was performed to exclude redundant variables in 50 train datasets (MASS package, version 7.3–54; stats package, version 4.0.5). Left main stenting and bifurcation stenting were unselected due to their high exclusion frequency (S2 Fig). Then significant collinearity was detected between the number of stenting vessels and total stent length using the variance inflation factor (mctest package, version 1.3.1). We unselected the number of stenting vessels to alleviate the collinearity, because the total stent length had high scores in the variable importance analysis, and it was not excluded once by stepwise AIC method.

A total of 12 variables were finally selected. Patient age, number of stenotic vessels, total stent length, and minimum stent diameter were input as continuous variables. CKD stage was input as ordered categorical variables. The male gender, smoking history, stenting for ACS, diabetes, hypertension, dyslipidemia, and complex lesions (type B2 and C) were input as binary variables. For RF model development using 10-fold cross-validation, the mtry and ntree were tuned each time.

## Model performance and interpretation

The predictive performance was evaluated using the test datasets. The area under the curve (AUC) of receiver operating characteristic (ROC) and precision-recall (PR) curves were calculated (pROC package, version 1.17.0). The sensitivity, specificity, positive predictive value, negative predictive value, detection rate, detection prevalence, F1 score, and accuracy at the cutoff thresholds of 0.8 sensitivity were evaluated using a confusion matrix. The variable importance in the RF model was assessed by CPI, MDA, and MDGini.

## Test model robustness

The model robustness was tested by removing outliers from the total data. We initialized a logistic regression model using the total data by inputting the 12 variables. The outliers were detected by the Cooks distance (stats package, version 4.0.5). Then we sequentially removed patients with more than 8 to 4 times of mean Cooks distance (mCD) from the study population, and reperformed the model development and validation in the 50 paired train and test datasets (S3 Fig). The ROC curves were analyzed to evaluate the model accuracy.

## Statistical analysis

For baseline characteristics, continuous variables were presented as mean ± standard deviation (SD). The Student's t-test was used for comparison if normally distributed; otherwise, the Mann-Whitney U test was used. Categorical variables were presented as frequency (percentage), and comparison was performed using the Chi-square test. The predictive metrics were presented as mean ± SD (minimum ~ maximum). The AUC-ROC were compared using the DeLong test. The AUC-PR were compared by values as no established statistical method. The accuracy, F1, sensitivity, specificity, PPV, and NPV in 50 test datasets were compared using paired Student's t-test. A two-tailed P value of less than 0.05 was considered statistically significant. All the statistical analysis was performed by R software 4.0.5.

# Results

## Study population

An overall number of 1501 patients with 279 (18.6%) ISR events were included in the study. Patient baseline characteristics were shown in Table 1. The age and male gender distributions between control patients and those with ISR were similar (control: age 63.6 ± 10.5 years, 76.2% male; ISR: age 65.7 ± 9.6 years, 78.9% male). The prevalence of smoking history, hypertension, dyslipidemia, diabetes, CKD stage, ACS, number of stenotic vessels, and number of stenting vessels were significantly higher in patients with ISR than those in control patients. The patients with ISR received stents with smaller minimum diameters and longer total lengths than the control patients.

**Table 1. Baseline characteristics.**

| | Control n = 1222 | ISR n = 279 | P value |
|---|---|---|---|
| Male gender | 931 (76.2) | 220 (78.9) | 0.383 |
| Age, yrs | 63.6 ± 10.5 | 65.7 ± 9.6 | 0.001 |
| Bodyweight, kg | 70.6 ± 11.5 | 71.5 ± 11.1 | 0.23 |
| Smoking history | 413 (33.8) | 118 (42.3) | 0.009 |
| Hypertension | 706 (57.8) | 182 (65.2) | 0.026 |
| Dyslipidemia | 379 (31.0) | 115 (41.2) | 0.001 |
| Diabetes | 433 (35.4) | 171 (61.3) | < 0.001 |
| CKD stage | | | |
| I or II | 944 (77.3) | 164 (58.8) | < 0.001 |
| III | 223 (18.2) | 69 (24.7) | |
| IV | 43 (3.5) | 30 (10.8) | |
| V | 12 (1.0) | 16 (5.7) | |
| LVEF, % | 50.8 ± 8.0 | 49.0 ± 8.4 | 0.002 |
| DAPT | 1198 (98.0) | 269 (96.4) | 0.101 |
| Statins | 1203 (98.4) | 274 (98.2) | 0.775 |
| ACEI/ARB | 1117 (91.4) | 258 (92.5) | 0.562 |
| β-blocker | 1075 (88.0) | 247 (88.5) | 0.794 |
| ACS | 948 (77.6) | 232 (83.2) | 0.049 |
| Number of stenotic vessels | | | |
| 1 vessel | 615 (50.3) | 100 (35.8) | < 0.001 |
| 2 vessels | 392 (32.1) | 97 (34.8) | |
| 3 vessels | 215 (17.6) | 82 (29.4) | |
| Left main stenosis | 48 (3.9) | 22 (7.9) | 0.008 |
| Number of stenting vessels | | | |
| 1 vessel | 959 (78.5) | 158 (56.6) | < 0.001 |
| 2 vessels | 229 (18.7) | 100 (35.8) | |
| 3 vessels | 34 (2.8) | 21 (7.5) | |
| Left main stenting | 48 (3.9) | 22 (7.9) | 0.008 |
| Bifurcation stenting | 33 (2.7) | 14 (5) | 0.07 |
| Complex lesion (type B2 and C) | 500 (40.9) | 201 (72.0) | < 0.001 |
| Minimum stent diameter, mm | 3.0 ± 0.4 | 2.7 ± 0.3 | < 0.001 |
| Total stent length, mm | 43.0 ± 21.6 | 59.7 ± 27.4 | < 0.001 |

Values are n (%) or mean ± SD.

ACEI = angiotensin-converting enzyme inhibitors; ACS = acute coronary syndrome; ARB = angiotensin receptor blockers; CKD = chronic kidney disease; DATP = dual antiplatelet therapy; LVEF = left ventricular ejection fraction.

## Model performance

The ROC and PR curves from 1 of the 50 test datasets are shown in Fig 2A and 2B. The RF models had an overall better predictive performance than the LR models (Table 2). The RF models predicted ISR with 0.45 ± 0.015 (0.000 ~ 0.075) larger AUC-ROC than LR models [0.829 ± 0.025 (0.783 ~ 0.880) vs. 0.784 ± 0.027 (0.722 ~ 0.835)]. The RF models had significantly larger AUC-ROC than the LR models in 29 of the 50 test datasets (Fig 2C). The AUC-PR was also larger in the RF model than that of LR model in 49 of the 50 test datasets (Fig 2D). The predictive metrics were assessed in the test datasets using the cutoff threshold of 0.8 sensitivity. The sensitivity, NPV, and detection rate were similar, but the RF models had significantly higher specificity, PPV, F1 score, accuracy, and lower detection prevalence than

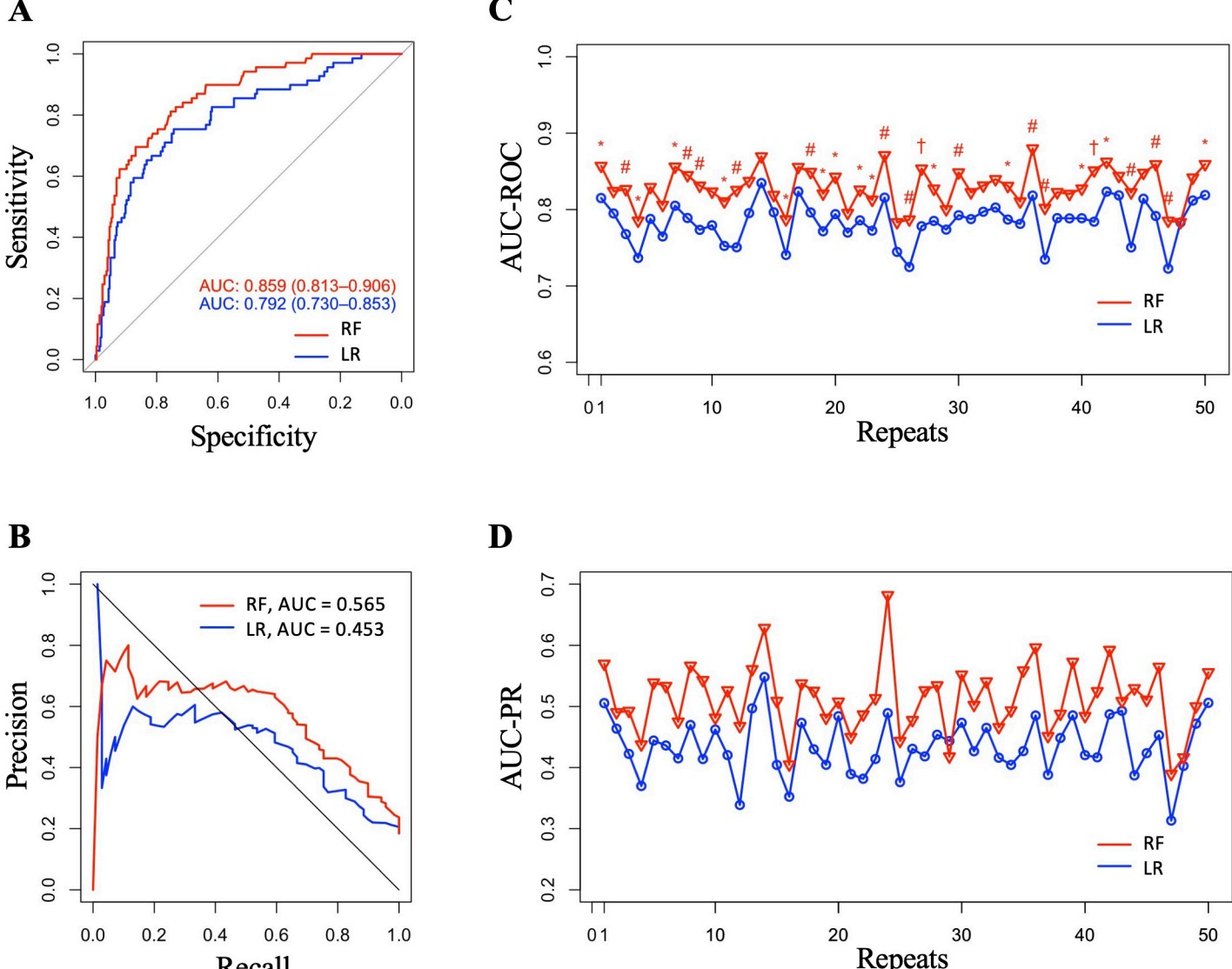

**Fig 2. Analysis of the ROC and PR curves.** The representative ROC (A) and PR curve (B) from 1 of the 50 test datasets are shown. The AUC-ROC (C) and AUC-PR (D) in the 50 test datasets were presented. The X-axis denotes each test dataset. The Y-axis denotes the value of AUC-ROC or AUC-PR. The AUC-ROCs between the RF and LR models were compared by the DeLong test. *P < 0.05, #P < 0.01, †P < 0.001. **Abbreviations:** AUC = area under the curve; LR = logistic regression; RF = random forest; ROC = receiver operating characteristic; PR = precision-recall.

LR in the majority of the test datasets. In general, the RF model predicted approximately 25% less false positive than the LR with similar sensitivity of 80%.

### Model robustness

The AUC-ROC significantly increased in both RF and LR models as the outliers were sequentially removed (Table 3). After the patients with more than 4 time of mCD were removed, the AUC-ROC was comparable between the RF and LR models [0.918 ± 0.016 (0.885 ~ 0.950) vs. 0.915 ± 0.016 (0.878 ~ 0.945)].

### Variable importance

The CPI, MDA, and MDGini were calculated from the 50 RF models (Fig 3). Although the results were discordant, the total stent length and minimum stent diameter ranked among the most important features for predicting ISR.

## Discussion

### Major findings

In the present study, using 50 random splits of paired train and test datasets, we found that the RF model was more robust and showed stable superiority in predicting angiographic ISR compared to the LR model. The total stent length and minimum stent diameter were the most important features for predicting ISR in the RF model.

### Comparison of the models

A model is considered to be robust if its accuracy is less affected by the outliers in the train dataset [15]. The robustness is usually tested by injecting outliers into the data. In our study, the multivariate outliers were the control patients with high ISR probability and the ISR patients who had low ISR probability. To test model robustness, we tailored the data by removing the patients with more than 4 times of mCD (S4 Fig). Then the LR model had comparable accuracy to the RF model. The result reversely provided the evidence that RF was more robust

**Table 2. The predictive performance between the RF and LR models.**

|  | Random forest | Logistic regression | Difference[a] | P value |
|---|---|---|---|---|
| AUC-ROC | 0.829 ± 0.025 (0.783 ~ 0.880) | 0.784 ± 0.027 (0.722 ~ 0.835) | 0.045 ± 0.015 (0.000 ~ 0.075) | 29/50[b] |
| AUC-PR | 0.512 ± 0.056 (0.389 ~ 0.682) | 0.435 ± 0.047 (0.313 ~ 0.548) | 0.077 ± 0.038 (-0.025 ~ 0.193) | NA |
| Sensitivity | 0.801 ± 0.057 (0.667 ~ 0.899) | 0.793 ± 0.062 (0.623 ~ 0.899) | 0.007 ± 0.053 (-0.116 ~ 0.101) | 0.335[c] |
| Specificity | 0.717 ± 0.031 (0.652 ~ 0.770) | 0.623 ± 0.033 (0.561 ~ 0.692) | 0.094 ± 0.035 (0.023 ~ 0.180) | < 0.001[c] |
| PPV | 0.392 ± 0.026 (0.333 ~ 0.444) | 0.323 ± 0.019 (0.272 ~ 0.362) | 0.069 ± 0.023 (0.016 ~ 0.127) | < 0.001[c] |
| NPV | 0.941 ± 0.015 (0.910 ~ 0.969) | 0.931 ± 0.018 (0.887 ~ 0.964) | 0.010 ± 0.015 (-0.026 ~ 0.036) | < 0.001[c] |
| Detection rate | 0.148 ± 0.011 (0.123 ~ 0.166) | 0.146 ± 0.011 (0.115 ~ 0.166) | 0.001 ± 0.010 (-0.021 ~ 0.019) | 0.335[c] |
| Detection prevalence | 0.378 ± 0.030 (0.316 ~ 0.439) | 0.454 ± 0.034 (0.366 ~ 0.524) | -0.076 ± 0.034 (-0.160 ~ -0.003) | < 0.001[c] |
| F1 score | 0.525 ± 0.029 (0.465 ~ 0.581) | 0.459 ± 0.025 (0.388 ~ 0.506) | 0.067 ± 0.025 (0.002 ~ 0.114) | < 0.001[c] |
| Accuracy | 0.759 ± 0.027 (0.705 ~ 0.811) | 0.708 ± 0.027 (0.634 ~ 0.758) | 0.051 ± 0.024 (-0.014 ~ 0.094) | < 0.001[c] |

Values are mean ± SD (minimum ~ maximum) from the 50 random test datasets.

[a] The value of random forest minus the value of logistic regression from each test dataset.

[b] DeLong test was used. P value less than 0.05 was revealed in 29 of the 50 test datasets.

[c] Paired student's T test was used.

AUC = area under the curve; NPV = negative predictive value; PPV = positive predictive value; PR = precision-recall; ROC = receiver operating characteristic.

**Table 3. Robustness test by sequentially removing the outliers.**

| | AUC-ROC | | | P < 0.05[b] |
| --- | --- | --- | --- | --- |
| | **Random forest** | **Logistic regression** | **Difference[a]** | |
| Total data control = 1222, ISR = 279 | 0.829 ± 0.025 (0.783 ~ 0.880) | 0.784 ± 0.027 (0.722 ~ 0.835) | 0.045 ± 0.015 (0.000 ~ 0.075) | 29/50[c] |
| Removal of the outliers with | | | | |
| > 8 times of mCD control = 1217, ISR = 266 | 0.836 ± 0.021 (0.786 ~ 0.875) | 0.801 ± 0.028 (0.734 ~ 0.871) | 0.035 ± 0.017 (-0.002 ~ 0.066) | 21/50[c] |
| > 7 times of mCD control = 1215, ISR = 255 | 0.845 ± 0.021 (0.813 ~ 0.897) | 0.815 ± 0.024 (0.775 ~ 0.864) | 0.030 ± 0.016 (-0.004 ~ 0.071) | 13/50[c] |
| > 6 times of mCD control = 1212, ISR = 236 | 0.872 ± 0.023 (0.825 ~ 0.914) | 0.850 ± 0.021 (0.806 ~ 0.889) | 0.021 ± 0.016 (-0.021 ~ 0.057) | 14/50[c] |
| > 5 times of mCD control = 1203, ISR = 204 | 0.900 ± 0.017 (0.859 ~ 0.932) | 0.886 ± 0.016 (0.853 ~ 0.923) | 0.014 ± 0.013 (-0.017 ~ 0.040) | 3/50[c] |
| > 4 times of mCD control = 1200, ISR = 173 | 0.918 ± 0.016 (0.885 ~ 0.950) | 0.915 ± 0.016 (0.878 ~ 0.945) | 0.003 ± 0.010 (-0.018 ~ 0.021) | 0/50[c] |

Values are mean ± SD (minimum ~ maximum) from the 50 test datasets.

[a] Value of random forest minus value of logistic regression.

[b] DeLong test was used.

[c] P value less than 0.05 was revealed in no. of the 50 test datasets.

mCD = mean Cooks distance.

than LR. However, the cutoff threshold of outlier is arbitrary, and the multivariate outlier is associated with the study population and variables. In addition, removing outliers to achieve higher accuracy is not feasible in prospective studies in which the outcome is unknown until observed.

Overfitting is the concept that a prediction model fits well with train datasets, but does not predict accurately with unseen test datasets. One of LR's limitations is when multiple variables with correlations are included, serious deviation would be generated and lead to overfitting [5]. The predictors of ISR are of multiple correlation and interaction [11]. Patients with diabetes are associated with renal dysfunction, dyslipidemia, long lesion, and smaller vascular diameter [16–18]. RF is an ensemble-based ML algorithm that uses multiple de-correlated decision trees to make a prediction. The tree-based model can be resistant to correlative variables [19].

## Important features

RF is not merely a black-box as other ML algorithms. We calculated the CPI, MDA, and MDGini to assess the importance of the variables. The CPI is considered to be more stable and reliable than the others [14]. The total stent length and minimum stent diameter ranked the most important features. Longer total stent length could imply a more complex vascular morphology and stenting approaches, such as multivessel disease, diffuse lesion, side branch, and bifurcate technique. Calcific lesion is associated with long lesion, older age, diabetes, and renal dysfunction. It is the major cause of stent under expansion and malapposition, which subsequently lead to ISR [20]. Longer total stent length could also correlate to smaller stent diameter in a diffuse lesion since more distal vessels could be targeted for stenting. A large-scale trial using intravascular ultrasound (IVUS) showed the cutoff of minimal stent area for prediction angiographic ISR was 5.3mm$^2$ ~ 5.7mm$^2$ [21]. By transforming to diameter, it was 2.6mm ~ 2.7mm, indicating a significantly higher ISR rate while implanting stents with diameters less than 2.75mm even in the absence of under expansion. Neoatherosclerosis is an important pathological characteristic of ISR in the second-generation DES era [22]. It can be accelerated atherosclerosis due to incomplete endothelialization, disrupted endothelial function, and excessive uptake of circulating lipid [23]. Smoking, hypertension, diabetes, CKD, and multivessel disease are associated with impaired endothelial function, and dyslipidemia contributes

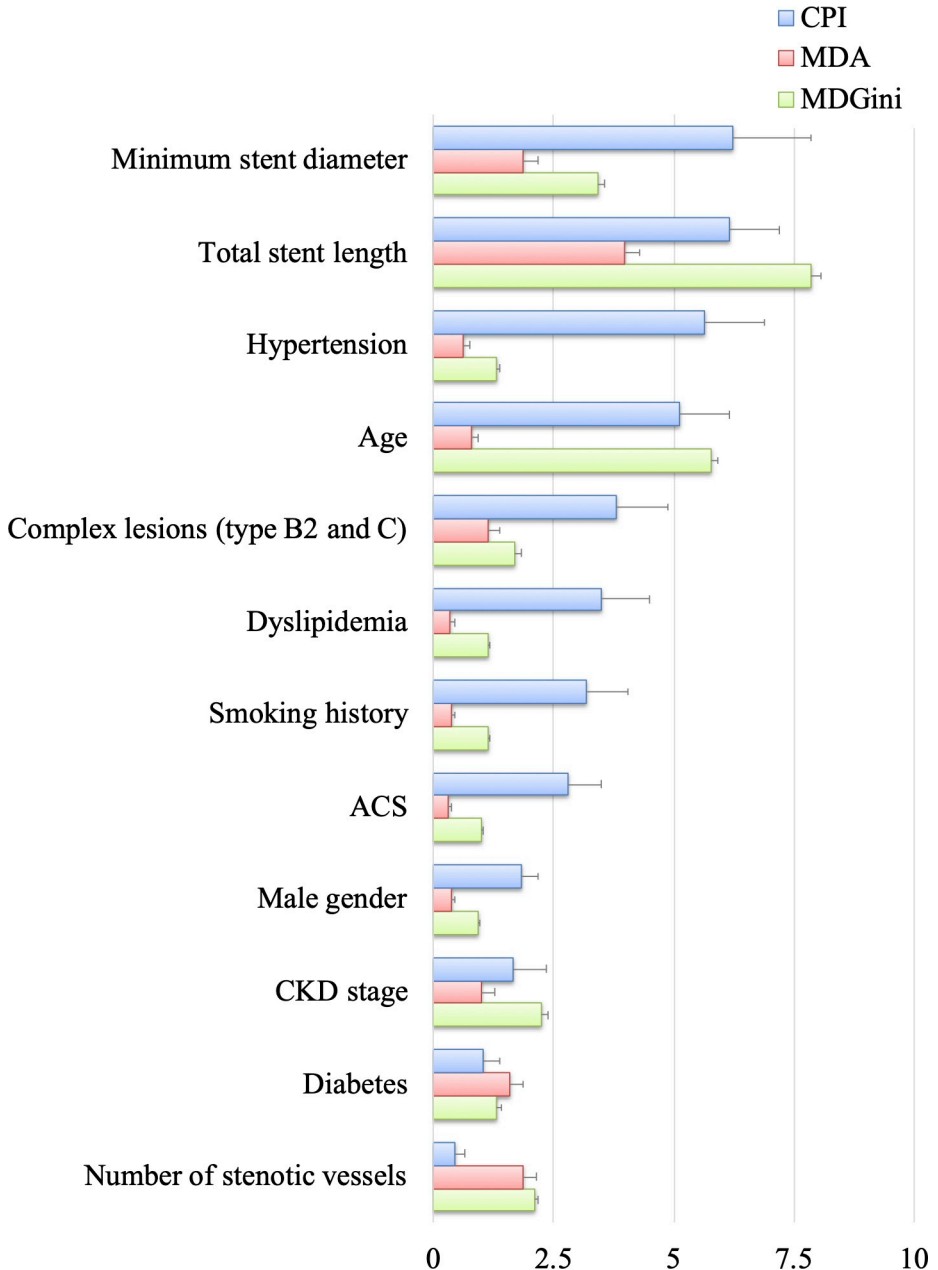

**Fig 3. Importance of the features.** The CPI, MDA, and MDGini of the variables. Sort by descending CPI. Values were all scaled to 0 ~10 for presentation. The higher the value, the more important the variable is. **Abbreviations:** ACS = acute coronary syndrome, CKD = chronic kidney disease, CPI = conditional permutation importance, MDA = mean decrease accuracy, MDGini = mean decrease Gini.

to a higher level of blood triglyceride and cholesterol. Stenting for ACS indicated stenting on unstable lesions that is a significant risk factor of neoatherosclerosis [22].

## Discordance

Jesús et al. compared 6 ML algorithms with 3 traditional risk score systems in predicting ISR using the data containing 263 patients from the GRACIA-3 trial [8]. A total of 68

variables, including bare-metal and DES stent, were screened for model development. Using the 10-fold cross-validation method, the RF and LR were among the models with the highest accuracy, but no statistically significant difference was revealed due to the small sample size (AUC-ROC: power = 0.218) and the possibility of noise variables [19]. We conducted the comparison of RF and LR in a larger retrospective patient cohort. Our study population consisted of patients receiving 2$^{nd}$ generation DES on de novo atherosclerotic lesions with worse renal function, multivessel disease, complex lesion, and small vessel diameter. It would therefore better fit the routine clinical practice in developing countries where coronary artery bypass graft surgery is not widely applicable. By randomly splitting the data into paired train and test datasets with 50 repeats, we found that the RF models all had larger AUC-ROC than the LR models. As the difference of sensitivity between the RF and LR model oscillated around zero, the differences of specificity, PPV, and F1 score were all above zero, indicating that the stable and better performance of the RF models is not by chance (S5 Fig). Cui et al. reported that 6 plasma metabolites can be used to predict ISR with a very high accuracy of 0.93 [24]. However, the metabolites are not routinely measured by mass spectrum in routine clinical practice, and their predictions were made after coronary stenting. Our model provides prediction before coronary stenting based on the variables obtained from daily practice, QCA, and stenting strategy.

## Clinical implication

Intravascular imaging modalities enable the ability to optimize PCI strategy and precise stenting [25]. The IVUS was associated with a 40% reduction in target vessel revascularization compared to angiographic guidance [26]. In our study, the intravascular imaging devices were only employed in less than 3% of the patients due to increased expense. The RF model predicted ISR with similar sensitivity of 80% but an average of 0.094 higher specificity than the LR model, reducing 25% of false positives. If the models had been used to identify patients with a high risk of ISR for employing intravascular imaging, close follow up and considering alternative therapy, the RF could have been more cost-effective than the LR by decreasing 25% expense with similar reductions in ISR and target vessel revascularization.

## Limitations

First, the current study is limited by its retrospective and single-center nature. The indications for repeat QCA included newly onset chest discomfort, prior high-risk PCI, and ischemic findings in non-invasive testing. The patients could be a self-selected high-risk group, questioning the validation in external and prospective cohorts. Further model generalization including less biased observational cohorts is required. Second, gene polymorphisms, blood biomarkers, intravascular imaging, coronary calcification, and PCI procedures which have been reported to be risk factors of ISR were not included in our models. Further feature selection for a more generalized model with better predictive performance is an ongoing work by our team. Third, the QCA was reassessed 9 ~ 18 months after initial coronary stenting. Some control patients who had reassessment of QCA early at 9 months may have diagnosed ISR if QCA was reassessed late to 18 months. This bias could result in an underestimation of the accuracy of the predictive models. Forth, the best cutoff threshold is unknown. Identifying more patients who will develop ISR with acceptable specificity is the rationale that 0.8 of sensitivity is used in our study. Finally, the RF and LR algorithm are limited in inputting coronary imaging data. The QCA could miss important features that are difficult to quantify. The convolutional neural network has the advantage of fully utilizing imaging data and could play a key role in future studies.

## Conclusions

Using the variables obtained from patient characteristics and QCA, we developed an ML model using RF to predict angiographic ISR in the retrospective cohort of patients who had initial coronary stenting for 9 ~ 18 months. The robust RF model improved predictive performance as compared with the traditional LR model and could help clinical decisions for coronary stenting with higher cost-effectiveness.

## Supporting information

**S1 Fig. Variable selection in the RF algorithm.** The conditional permutation importance, mean decrease accuracy, and mean decrease Gini of the 15 variables. Order as descending CPI value. The higher the value, the more important the variable is. **Abbreviations:** ACS = acute coronary syndrome, CKD = chronic kidney disease.
(TIF)

**S2 Fig. Variable exclusion frequency in the LR model by the stepwise AIC method.** The exclusion frequency was counted from the 50 LR models. The exclusion frequency is denoted at the top of each column. Order as ascending exclusion frequency. The higher the frequency, the variable less influenced the LR model. **Abbreviations:** ACS = acute coronary syndrome, CKD = chronic kidney disease.
(TIF)

**S3 Fig. The workflow of robustness test. Abbreviations:** ISR = intra-stent restenosis, ROC = receiver operating characteristic.
(TIF)

**S4 Fig. The Cooks distance and probability of ISR.** The Cooks distances among the study population (A). The X-axis denotes each patient. The Y-axis denotes the Cooks distance of each patient in ascending order. The blue solid line denotes the mCD. The red dashed line denotes the threshold of 4 times of mCD. The histogram of the probability of ISR of the study population (B) and that after removal of the patients with more than 4 times of mCD (C). The X-axis ranges from 0 to 1, denoting the probability of ISR. The Y-axis denotes the patient frequency of the probabilities. **Abbreviations:** ISR = in-stent restenosis; mCD = mean Cooks distance.
(TIF)

**S5 Fig. Difference of the metrics between the RF and LR models.** The difference was calculated by subtracting the value of the LR model from that of the RF model in each test dataset. The difference of sensitivity oscillated around zero in the 50 test datasets. However, the differences in specificity, PPV, and F1 scores were all above zero, indicating that the RF models had higher specificity, PPV, and F1 scores than the LR models under similar sensitivity. **Abbreviations:** LR = logistic regression; RF = random forest.
(TIF)

**S1 File. Minimal data set.**
(XLSX)

## Acknowledgments

We thank Jason Tri from Mayo Clinic for his dedicated help in writing the manuscript.

## Author Contributions

**Conceptualization:** Zhi Jiang.

**Data curation:** Longhai Tian, Wei Liu, Bo Song, Chao Xue, Tianzong Li, Jin Chen.

**Formal analysis:** Zhi Jiang, Longhai Tian.

**Funding acquisition:** Zhi Jiang, Wei Liu, Fang Wei.

**Investigation:** Longhai Tian, Wei Liu, Bo Song, Chao Xue, Tianzong Li, Jin Chen.

**Methodology:** Zhi Jiang.

**Project administration:** Fang Wei.

**Resources:** Longhai Tian, Wei Liu, Bo Song, Chao Xue, Tianzong Li, Jin Chen.

**Software:** Zhi Jiang, Tianzong Li.

**Supervision:** Fang Wei.

**Validation:** Longhai Tian, Wei Liu.

**Visualization:** Longhai Tian, Wei Liu, Tianzong Li.

**Writing – original draft:** Zhi Jiang.

**Writing – review & editing:** Zhi Jiang, Longhai Tian, Wei Liu, Fang Wei.

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
