## [Decision Letter · Decision Letter 0]

26 Dec 2021

PONE-D-21-30474Random forest vs. logistic regression: predicting angiographic in-stent restenosis after second-generation drug-eluting stent implantationPLOS ONE

Dear Dr. Fang,

Thank you for submitting your manuscript to PLOS ONE. After careful consideration, we feel that it has merit but does not fully meet PLOS ONE’s publication criteria as it currently stands. Therefore, we invite you to submit a revised version of the manuscript that addresses the points raised during the review process.

We look forward to receiving your revised manuscript.

Kind regards,

Ankur Sethi

Academic Editor

PLOS ONE

Journal Requirements:

2. Please update your Methods section to state that the requirement for informed participant consent was waived by your ethics committee due to the fact that data was anonymized.

FW recieved the Clinical Research Center Project of Department of Science and Technology of Guizhou Province [NO.(2017)5405]; ZJ recieved the Guizhou Provincial High-level Innovative Talents Project (GZSYQCC[2015]006); WL recieved the Guizhou Provincial Science and Technology Foundation (No.[2019]1197); FW recieved the Guizhou Provincial Science and Technology Social Development Project (No.[2018]2794).

Additional Editor Comments:

Issues –

Page 2 line 43 – I think the mortality data is controversial for acute coronary syndrome. May be authors meant acute ST elevation myocardial infarction?

Page 3 line 44 – the reference 1 cites IFR-SWEDHEART trial which compared FFR to iFR and probably not an appropriate reference for improvement in outcomes after PCI for ACS.

Authors selected 1501 out of 2508 patients (59%) patients as they underwent a repeat coronary angiogram within 9-18 months of the index procedure. First, this misses out on cases of instent restenosis in rest 41% patients who did not receive coronary angiogram for various reasons including – death, no symptoms, or other medical reasons. Secondly, the reasons for the patients who were scheduled for repeat angiogram is unclear. Was it for staged intervention, new symptoms, or ACS? Therefore, these patients may be self-selected high risk group who received an unplanned angiogram compared to 41% who did not, questioning the validity and applicability of these models to all comers.

“We excluded patients receiving stents by operators from other hospital” These patients were treated at Guizhou Provincial hospital?

How many patients were excluded due missing data, and use of stent other than 2nd generation stents?

Was there evidence of ischemia based on non-invasive testing in the territory of vessel with in-stent restenosis?

Were patients with PCI of saphenous venous graft included?

It may be of value to include severe calcification and/or use of atherectomy as predictor of in-stent restenosis.

Would author consider use of intra-coronary imaging as a predictor for in-stent restenosis. What was the proportion of patients with imaging in two groups?

Minor issues –

Can authors briefly discuss how should there results be used in a clinical meaningful way to predict and/or prevent restenosis ?

Reviewers' comments:

Reviewer's Responses to Questions

**Comments to the Author**

1. Is the manuscript technically sound, and do the data support the conclusions?

Reviewer #1: Yes

2. Has the statistical analysis been performed appropriately and rigorously? 

Reviewer #1: Yes

3. Have the authors made all data underlying the findings in their manuscript fully available?

Reviewer #1: Yes

4. Is the manuscript presented in an intelligible fashion and written in standard English?

Reviewer #1: Yes

5. Review Comments to the Author

Reviewer #1: Wei et al presents a well-written and timely manuscript analyzing the utility of RF versus LR in predicting ISR after second generation DES implantation. RF may provide added benefits over LR in clinical practice given its ability to identify more complex feature patterns and provide better accuracy as compared to LR. Additionally, RF can also provide feature importance when predicting a specific outcome, whereas LR cannot. In the specific case of ISR as presented by the authors, RF may provide specific clinical prediction benefit over LR given the nature of significant interaction and multiple correlation between the predictors of ISR. LR analysis will likely provide less accuracy in predicting outcomes as seen in the manuscript presented by Wei et al due to these complex interactions between predictors of ISR and argues for the utilization of RF over LR in similar clinical scenarios where predictors of a specific outcome may have multiple correlation, and argues that RF may have important use in clinical practice over more traditional statistical analysis methods like LR. The importance of prospective trials in assessing the clinical utility of RF needs to be emphasized, and the retrospective nature of the manuscript is an important limitation. Outliers will not be readily identifiable in prospective studies if the study outcome is not known, which may limit the accuracy and potential clinical application of RF models. The authors should address in the manuscript whether any prospective validation studies utilizing RF in observational cohorts have been previously published and the outcomes of these studies in reference to their clinical utility.

6. PLOS authors have the option to publish the peer review history of their article (what does this mean?). If published, this will include your full peer review and any attached files.

Reviewer #1: No

---

## [Author Response · Author response to Decision Letter 0]

16 Jan 2022

Journal Requirements:

1. Please ensure that your manuscript meets PLOS ONE's style requirements. 

We have carefully checked the style. We ensure our manuscript meets PLOS ONE’s style requirement.

2. Please update your Methods section to state that the requirement for informed participant consent was waived by your ethics committee due to the fact that data was anonymized.

We have included the statement in Page 5 line 74-75.

3. Thank you for stating the financial disclosure. Please state what role the funders took in the study. Please include this amended Role of Funder statement in your cover letter; we will change the online submission form on your behalf.

We have included the funder statement in our cover letter. They took no role in the study.

4. In your Data Availability statement, you have not specified where the minimal data set underlying the results described in your manuscript can be found.Upon re-submitting your revised manuscript, please upload your study’s minimal underlying data set as either Supporting Information files or to a stable, public repository and include the relevant URLs, DOIs, or accession numbers within your revised cover letter. 

We have uploaded the minimal data set as Supporting Information files. We added S1 File in page 23 line 460.

Additional Editor Comments:

Issues

1. Page 2 line 43 – I think the mortality data is controversial for acute coronary syndrome. May be authors meant acute ST elevation myocardial infarction?

Thank you for pointing out the controversy. We have revised ACS to ST-segment elevation myocardial infarction. 

2. Page 3 line 44 – the reference 1 cites IFR-SWEDHEART trial which compared FFR to iFR and probably not an appropriate reference for improvement in outcomes after PCI for ACS.

Thank you for pointing out the inappropriate reference. We have revised Reference 1 to the 2018 ESC guidelines on myocardial revascularization.

3. Authors selected 1501 out of 2508 patients (59%) patients as they underwent a repeat coronary angiogram within 9-18 months of the index procedure. First, this misses out on cases of instent restenosis in rest 41% patients who did not receive coronary angiogram for various reasons including – death, no symptoms, or other medical reasons. Secondly, the reasons for the patients who were scheduled for repeat angiogram is unclear. Was it for staged intervention, new symptoms, or ACS? Therefore, these patients may be self-selected high risk group who received an unplanned angiogram compared to 41% who did not, questioning the validity and applicability of these models to all comers.

We notice that the data source was not expressed clearly. All the 2508 patients had a repeat coronary artery angiography (CAG) and 360(14.4%) of them were diagnosed in-stent restenosis (ISR). Among the 1007(41%) patients who were excluded, 967 had prior coronary stenting in other hospitals, 16 had stents other than 2nd generation drug-eluting stents, and 24 had missing clinical data. We have revised the expression in Page 5 line 77-82. The numbers have been added in Fig 1 patient flow. 

We agree with the editor that the patients could be a self-selected high-risk group, and the external validation of these models is questioned. A more generalized model would be more meaningful in clinical practice, but selection bias and external validation might always be the limitation. Model generalization is our ongoing work. We have added the limitation in Page 17 line 298-302.

The most frequent indications for repeat CAG included newly onset chest discomfort, prior high-risk PCI, and ischemic findings at non-invasive testing. It is difficult to trace back all the indications due to the retrospective nature. In the study, our primary goal was to compare the predictive performance between two machine-learning algorithms. We revealed that the resistance to outliers resulted in the better predictive performance of the random forest than the logistic regression. We respectfully suggest that further investigation of the indications would not add significantly to our major findings and conclusions. 

4. “We excluded patients receiving stents by operators from other hospital” These patients were treated at Guizhou Provincial hospital?

These patients had reassessment by CAG in the Guizhou Provincial Hospital, but their prior coronary stenting was performed in other hospitals. We have revised the expression in Page 5 line 77-82.

5. How many patients were excluded due missing data, and use of stent other than 2nd generation stents?

24 patients were excluded due to missing data. The numbers have been shown in Fig 1 and included in Page 5 line 80-82.

6. Was there evidence of ischemia based on non-invasive testing in the territory of vessel with in-stent restenosis?

There was ischemic evidence in the patients who had ISR. But it is difficult to trace back all the non-invasive testing and ischemic territory due to its retrospective nature. The rationale of our machine-learning models was to predict ISR before coronary stenting using the variables from daily practice, CAG, and stenting strategy. The prediction could help identify patients with high-risk ISR for employing intravascular imaging in PCI procedures, close follow up or consideration of alternative therapy. We respectfully suggest that further correlating the non-invasive testing and ischemic territory would not add significantly to our major findings and conclusion.

7. Were patients with PCI of saphenous venous graft included?

CABG was far more less performed than PCI in our district due to patient preference. No patients with PCI of bypass grafts were included in the study. We have added the statement in Page 5 line 84.

8. It may be of value to include severe calcification and/or use of atherectomy as predictor of in-stent restenosis.

9. Would author consider use of intra-coronary imaging as a predictor for in-stent restenosis. What was the proportion of patients with imaging in two groups?

We would like to respond to issue 8&9 together. We agree with the editor that it is of value if calcification and/or use of a specific device are added to the predictors. The characteristics of calcification such as depth, thickness, angle, length, as well as the fracture of calcific plaque after stenting were significantly associated with the prognosis.[1–4] However, the intracoronary imaging devices were employed in less than 3% of the patients. We were not able to report detailed calcific characteristics (Page 17 line 294-296). We notice that the random forest and logistic regression algorithms are limited in utilizing the imaging data because quantifying imaging data into a group of variables could have lost important features. The emerging convolutional neural network could overcome the limitation by inputting the digital imaging data.[5] It could play a key role in guiding PCI and predicting clinical outcomes in future studies. We respectfully suggest that further including new features would not add significantly to our major findings and conclusion. The limitation has been added in Page 17 line 311-314.

References

1. Wang X, Matsumura M, Mintz GS, Lee T, Zhang W, Cao Y, et al. In Vivo Calcium Detection by Comparing Optical Coherence Tomography, Intravascular Ultrasound, and Angiography. JACC Cardiovasc Imaging. 2017;10: 869–879. doi:10.1016/j.jcmg.2017.05.014

2. Sharma SK, Vengrenyuk Y, Kini AS. IVUS, OCT, and Coronary Artery Calcification: Is There a Bone of Contention?∗. JACC: Cardiovascular Imaging. 2017;10: 880–882. doi:10.1016/j.jcmg.2017.06.008

3. Fujino A, Mintz GS, Lee T, Hoshino M, Usui E, Kanaji Y, et al. Predictors of Calcium Fracture Derived From Balloon Angioplasty and its Effect on Stent Expansion Assessed by Optical Coherence Tomography. JACC Cardiovasc Interv. 2018;11: 1015–1017. doi:10.1016/j.jcin.2018.02.004

4. Maejima N, Hibi K, Saka K, Akiyama E, Konishi M, Endo M, et al. Relationship Between Thickness of Calcium on Optical Coherence Tomography and Crack Formation After Balloon Dilatation in Calcified Plaque Requiring Rotational Atherectomy. Circ J. 2016;80: 1413–1419. doi:10.1253/circj.CJ-15-1059

5. Anwar SM, Majid M, Qayyum A, Awais M, Alnowami M, Khan MK. Medical Image Analysis using Convolutional Neural Networks: A Review. J Med Syst. 2018;42: 226. doi:10.1007/s10916-018-1088-1

Minor issues

1. Can authors briefly discuss how should there results be used in a clinical meaningful way to predict and/or prevent restenosis ?

Yes. We have added the discussion in Page 16 line 287-295.

Comments to the Author

Reviewer #1: Wei et al presents a well-written and timely manuscript analyzing the utility of RF versus LR in predicting ISR after second generation DES implantation. RF may provide added benefits over LR in clinical practice given its ability to identify more complex feature patterns and provide better accuracy as compared to LR. Additionally, RF can also provide feature importance when predicting a specific outcome, whereas LR cannot. In the specific case of ISR as presented by the authors, RF may provide specific clinical prediction benefit over LR given the nature of significant interaction and multiple correlation between the predictors of ISR. LR analysis will likely provide less accuracy in predicting outcomes as seen in the manuscript presented by Wei et al due to these complex interactions between predictors of ISR and argues for the utilization of RF over LR in similar clinical scenarios where predictors of a specific outcome may have multiple correlation, and argues that RF may have important use in clinical practice over more traditional statistical analysis methods like LR. 

1. The importance of prospective trials in assessing the clinical utility of RF needs to be emphasized, and the retrospective nature of the manuscript is an important limitation. 

We have added the limitation in Page 17 line 298-302.

2. Outliers will not be readily identifiable in prospective studies if the study outcome is not known, which may limit the accuracy and potential clinical application of RF models. 

We agree with the reviewer that removing outliers is not feasible in prospective studies. The process that sequentially removing the patients according to Cook’s distance was to reveal the mechanism that the random forest was superior in predictive accuracy than the logistic regression. We had the comment in Page 14 line 232-233.

3. The authors should address in the manuscript whether any prospective validation studies utilizing RF in observational cohorts have been previously published and the outcomes of these studies in reference to their clinical utility.

Sampedro-Gómez et al first trained and validated the machine-learning models to predict ISR using the cohort from the prospective randomized control trial (GRACIA-3). They revealed that the random forest had better predictive performance than the logistic regression, but the difference was not statistically significant. The study was introduced in Page 3 line 61-63, reference 8.

---

## [Decision Letter · Decision Letter 1]

9 May 2022

Random forest vs. logistic regression: predicting angiographic in-stent restenosis after second-generation drug-eluting stent implantation

PONE-D-21-30474R1

Dear Dr. Wei,

We’re pleased to inform you that your manuscript has been judged scientifically suitable for publication and will be formally accepted for publication once it meets all outstanding technical requirements.

Kind regards,

George Vousden

Deputy Editor-in-Chief

PLOS ONE

Additional Editor Comments (optional):

Reviewers' comments:

Reviewer's Responses to Questions

**Comments to the Author**

1. If the authors have adequately addressed your comments raised in a previous round of review and you feel that this manuscript is now acceptable for publication, you may indicate that here to bypass the “Comments to the Author” section, enter your conflict of interest statement in the “Confidential to Editor” section, and submit your "Accept" recommendation.

Reviewer #1: All comments have been addressed

2. Is the manuscript technically sound, and do the data support the conclusions?

Reviewer #1: Yes

3. Has the statistical analysis been performed appropriately and rigorously? 

Reviewer #1: Yes

4. Have the authors made all data underlying the findings in their manuscript fully available?

Reviewer #1: Yes

5. Is the manuscript presented in an intelligible fashion and written in standard English?

Reviewer #1: Yes

6. Review Comments to the Author

Reviewer #1: (No Response)

7. PLOS authors have the option to publish the peer review history of their article (what does this mean?). If published, this will include your full peer review and any attached files.

Reviewer #1: No

---

## [Editor Report · Acceptance letter]

12 May 2022

PONE-D-21-30474R1 

Random forest vs. logistic regression: predicting angiographic in-stent restenosis after second-generation drug-eluting stent implantation 

Dear Dr. Wei:

I'm pleased to inform you that your manuscript has been deemed suitable for publication in PLOS ONE. Congratulations! Your manuscript is now with our production department. 

Kind regards, 

on behalf of

Dr. George Vousden 

Staff Editor

PLOS ONE